# OX40L–OX40 Signaling in Atopic Dermatitis

**DOI:** 10.3390/jcm10122578

**Published:** 2021-06-11

**Authors:** Masutaka Furue, Mihoko Furue

**Affiliations:** 1Department of Dermatology, Kyushu University, Higashiku, Fukuoka 812-8582, Japan; 21-19-20 Momochi, Sawara-ku, Fukuoka 814-0006, Japan; furue1@jcom.home.ne.jp

**Keywords:** OX40L, OX40, atopic dermatitis, Th1 cell, Th2 cell, Th17 cell, dendritic cell

## Abstract

OX40 is one of the co-stimulatory molecules expressed on T cells, and it is engaged by OX40L, primarily expressed on professional antigen-presenting cells such as dendritic cells. The OX40L–OX40 axis is involved in the sustained activation and expansion of effector T and effector memory T cells, but it is not active in naïve and resting memory T cells. Ligation of OX40 by OX40L accelerates both T helper 1 (Th1) and T helper 2 (Th2) effector cell differentiation. Recent therapeutic success in clinical trials highlights the importance of the OX40L–OX40 axis as a promising target for the treatment of atopic dermatitis.

## 1. Introduction

T cells are pivotal players in adaptive immune responses, exhibiting a wide range of activities including cytokine/chemokine secretion, B cell regulation, macrophage response regulation, and cytotoxicity against infected or neoplastic cells. Full T cell activation requires three key signals: engagement of T cell receptors by antigen peptide-bound major histocompatibility complex, ligation of co-stimulatory molecules, and support from cytokines [1]. In addition, recent studies have stressed a cardinal role of co-inhibitory molecules in regulating excessive T cell activation [2,3,4,5]. There are several co-stimulatory and co-inhibitory molecules, also known as immune checkpoint molecules (Figure 1). 

The therapeutic success of blocking antibodies for co-inhibitory molecules, such as programmed cell death 1, programmed death ligand 1, and cytotoxic T lymphocyte-associated protein 4, against melanoma and other cancers supports the multidirectional potential for targeting other immune checkpoint molecules in treating malignancies, autoimmune diseases, and allergic disorders [3,4,6,7,8]. However, only a few of molecules have been well characterized.

Immune checkpoint molecules are primarily classified into two groups: the immunoglobulin superfamily, and the tumor necrosis factor superfamily (TNFSF) and its receptors (TNFRSFs) [3,4,9] (Figure 1). OX40 (TNFRSF4, CD134) and its ligand OX40L (TNFSF4, CD252) are two such TNFSF/TNFRSF co-stimulatory immune checkpoint molecules [10,11]. In general, ligation of OX40 onto T cells by OX40L on antigen-presenting cells (APCs) facilitates the effector function of T cells [10,11]. OX40 expression can be induced in a variety of T cell subsets [10,11]. It is reported that the co-stimulatory molecules CD28 and CD40 regulate the early phases of T cell activation, whereas OX40 and CD30 are probably crucial for the late phases or long-lasting T cell responses [10,11]. Agonistic antibodies against OX40 induced favorable anti-tumor immune responses in preliminary clinical trials [9]. By contrast, antagonistic antibodies against OX40 produced favorable outcomes in the treatment of atopic dermatitis [12,13]. Therefore, the OX40–OX40L axis may be critical for promoting the type 2 immune deviation found in human atopic dermatitis [14,15].

## 2. OX40 Expression and Signaling in T Cells

OX40 is expressed primarily by activated T cells including CD4+ and CD8+ T cells; Th1, Th2, and Th17 cells; and Foxp3+CD4+ regulatory T cells (Tregs) [10,11]. It is not expressed by naïve CD4+ and CD8+ T cells, or most resting memory T cells. However, OX40 is rapidly expressed by these naïve and memory cells upon activation [16,17]. OX40 expression is induced and upregulated via the cross-linking of T cell receptors together with co-stimulation via CD28 (Figure 1). Its expression is enhanced and prolonged by IL-1, IL-2, IL-4, and TNF-α [10,11]. In various experimental autoimmune models, the OX40–OX40L axis enhances the effector function of CD4+ T cells, and its blockade ameliorates autoimmune inflammation [18]. Contrarily, the activation of OX40 by OX40L inhibits the expression of Foxp3 and the inhibitory function of Tregs [19,20,21]. OX40L is preferentially expressed in professional APCs, including dendritic cells, macrophages, and activated B cells [10,11]. Its expression is augmented by the activation of CD40 and Toll-like receptors [10,11]. Binding of OX40L results in the trimerization of OX40 monomers; recruitment of TRAF2, TRAF3, and TRAF5; and activation of NF-κB and NFAT pathways [10,11].

## 3. OX40–OX40L Axis in Type 1 and Type 2 Immune Responses

OX40 is expressed in activated effector T and effector memory T cells. Its stimulation promotes and sustains T helper 1 (Th1) and Th2 activation rather than B cell responses [10,11,22,23] (Figure 2). 

In a viral infection model, OX40 deficiency did not affect the production of virus-specific antibody production or virus-specific cytotoxic T cell response [22]. However, the proliferation of virus-specific Th cells and the production of interferon-γ (namely Th1 differentiation) are impaired in OX40-deficient mice [22]. Th1 differentiation was also attenuated in a systemic *Listeria monocytogenes* infection model of OX40-deficient mice. NK cell-derived interferon-γ upregulates OX40L expression in dendritic cells, and OX40L-expressing dendritic cells are critically involved in OX40+ Th1 cell expansion in this model [23]. Interestingly, OX40L-expressing group 3 innate lymphoid cells (ILC3s), but not dendritic cells, were required for full Th1 cell differentiation in an intestinal *Salmonella* infection model, suggesting that different APC population may be crucial for OX40L-dependent, site-specific inflammation [23]. 

Type 2 immune responses are pathogenic in allergic disorders such as atopic dermatitis and asthma [15,24,25]. In a murine asthma model using repeated ovalbumin administration, OX40 deficiency resulted in a significant reduction of airway hyperreactivity and Th2 responses such as IL-4 and IgE production [26]. Fms-like tyrosine kinase 3 is a cardinal molecule for generating functional dendritic cells [27]. Respiratory syncytial virus infection also induces Th2-prone respiratory inflammation [28]. In this model, OX40L+ ILC2 and OX40+ Th cells are involved in Th2 cell expansion [28]. The administration of small interference mRNA targeting OX40L alleviates ovalbumin-induced allergic rhinitis by inhibiting the Th2-prone immune response [29]. Dendritic cells expressing CD301b are known to preferentially induce Th2-deviated immune responses [30]. Repeated urinary tract infection activates Th2-biased responses, which are mediated by CD301b+ OX40L+ dendritic cells [31].

Thus, the interaction of OX40L on APCs with OX40 on Th cells promotes and sustains both Th1 and Th2 cell expansion. Meanwhile, OX40 signaling likely inhibits IL-17A production and Th17 differentiation in an experimental autoimmune encephalomyelitis model [32]. Ligation of OX40 by recombinant OX40L upregulates the expression of interferon-γ and IL-4 in human T cells stimulated with phytohemagglutinin, whereas it downregulates the production of IL-17A [33]. However, conflicting results have been reported in models of uveitis and intestinal inflammation, in which OX40 signaling supports Th17 cell differentiation in vivo [34,35]. Further investigation is warranted to clarify the role of OX40L–OX40 signaling in Th17 differentiation.

OX40 shares signaling pathways with its fellow TNFRSF member CD30 (TNFRSF8) [36,37,38]. Therefore, the CD30L–CD30 axis is known to exert synergistic effects with the OX40L–OX40 axis in both Th1 and Th2 differentiation [23,37,39].

## 4. OX40L–OX40 Axis in Atopic Dermatitis

Atopic dermatitis is characterized with Th2-biased immune responses, skin barrier dysfunction, and pruritus [15,24]. Th2-derived IL-13 and IL-4 decrease the expression of skin barrier-related proteins, such as filaggrin and loricrin, and cause barrier dysfunction [14]. In turn, barrier dysfunction accelerates Th2 immune deviation, which is attributable to pro-type 2 cytokines such as thymic stromal lymphopoietin (TSLP), IL-25, and IL-33 produced from barrier-disrupted epidermis [40,41,42,43]. 

TSLP stimulates murine dendritic cells to express OX40L, and OX40L-positive dendritic cells induce OX40-positive T-cell differentiation [44,45,46,47]. OX40-positive T cells include a large number of Th2 cells expressing IL-4, IL-5, and IL-13 [44,45,46,47]. Human dendritic cells treated with TSLP also tend to induce a Th2-cell population producing IL-4, IL-5, and IL-13 [43]. Of note, TSLP-induced atopic inflammation is prevented by OX40L-specific blocking antibody [48].

IL-25 transgenic mice exhibit an increased number of blood eosinophils; elevated serum IgE levels; and IL-4, IL-5, and IL-13 hyperproduction [49]. The intranasal administration of IL-25 induces pulmonary hypereosinophilia and enhances the expression of IL-4, IL-5, IL-13, and eotaxin [50]. IL-25 also stimulates dendritic cells to express OX40L, and results in Th2 differentiation [51]. 

IL-33 is a member of the IL-1 family that is produced by peripheral tissues and that induces type 2-dominant immune deviation [52]. It is overexpressed in keratinocytes derived from tape-stripped, barrier-disrupted epidermis [53]. IL-33 also stimulates ILC2s and dendritic cells to express OX40L, and upregulates Th2 differentiation [54,55]. IL-25 can also enhance OX40L expression in ILC2s, but its potency is lower than that of IL-33 [54]. Conversely, TSLP upregulates OX40L expression in dendritic cells but not in ILC2s [54]. 

The number of OX40L+ dendritic cells is higher in the lesional skin of patients with atopic dermatitis than in psoriatic and normal skin [40,56,57,58,59]. In the peripheral blood of patients with atopic dermatitis, OX40-expressing T cells are mostly compartmentalized in cutaneous lymphocyte-associated antigen-expressing CD45RO+ CD4+ T cells, namely skin-homing memory Th cells [60]. OX40+ dermal cells are also co-localized with OX40L+ cells, such as mast cells in the lesional skin of patients with atopic dermatitis [60]. These results stress that the OX40L–OX40 axis is probably an integral part of the pathogenic Th2 deviation crucial for atopic dermatitis. 

## 5. Therapeutic Intervention Targeting the OX40L–OX40 Axis in Atopic Dermatitis

GBR 830 is an antagonizing anti-OX40 antibody. In a phase 2a randomized, double-blinded, placebo-controlled study (NCT02683928), GBR 830 was administered intravenously twice on days 1 and 29 in adults with moderate-to-severe atopic dermatitis [12]. GBR 830 was safe and tolerable in 46 patients. GBR 830 decreased the expression of OX40 and OX40L in the lesional skin in association with significant normalization of histological and transcriptomic abnormalities. The treatment reduced Th2, Th1, and Th17/Th22 signatures in the lesional skin. GBR 830 also significantly decreased clinical symptoms of atopic dermatitis compared to the effects of placebo [12].

KHK4083 is an antagonistic anti-OX40 antibody with antibody-dependent cellular cytotoxicity [61]. In a phase 1 trial of KHK4083 enrolling 22 patients with atopic dermatitis, injection of this antibody on day 1, day 15 and day 29 ablated OX40-expressing cells and significantly attenuated the symptoms of atopic dermatitis compared to the effects of placebo [61]. Continued improvement in the Eczema Area and Severity Index and Investigator’s Global Assessment scores were observed throughout the study until day 155 [61]. A phase 2 clinical study of an anti-OX40 monoclonal antibody (KHK4083) in subjects with moderate-to-severe atopic dermatitis was recently completed (NCT03703102, *ClinialTrials.gov,* accessed on 10 June 2021) [13]. In this study, 274 patients with moderate to severe atopic dermatitis were enrolled. Although the final results are not published yet, this study also showed progressive improvement in efficacy by continuous KHK4083 administration beyond 16 w eeks and the potential for long-term sustained therapeutic effects after the completion of KHK4083 treatment [13]. These clinical studies further underpin the pathogenic implication of the OX40L–OX40 axis in atopic dermatitis.

## 6. Conclusions

The OX40L–OX40 axis is crucial for sustaining the effector function of T cells, especially Th1 and Th2 subpopulations. The expression of OX40 is confined to activated effector T and effector memory T cells. Therefore, therapeutic agents targeting OX40 do not affect naïve and resting memory T cells, which may explain the relatively safe properties of these agents. The recent clinical success of antagonistic or cytotoxic anti-OX40 antibodies in patients with atopic dermatitis underpins the potential efficacy of these agents for other allergic disorders. 

## Figures and Tables

**Figure 1 jcm-10-02578-f001:**
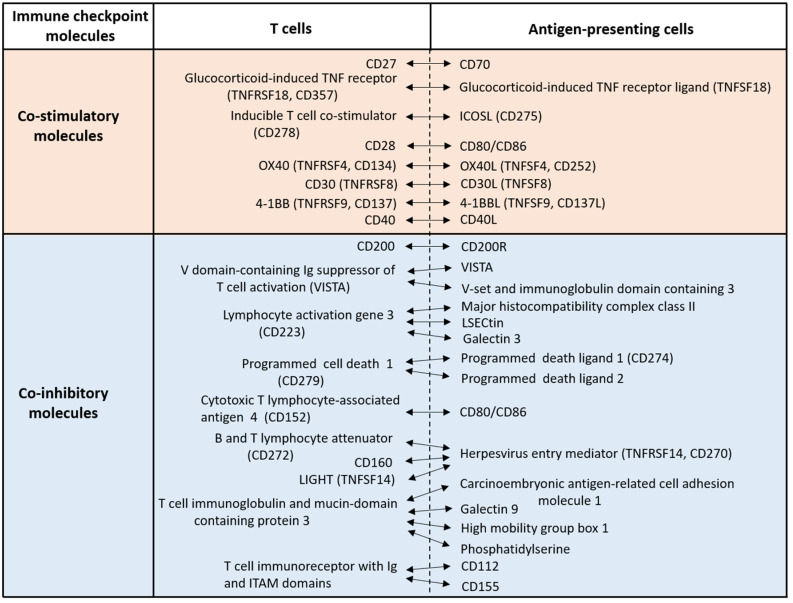
Representative immune checkpoint molecules. T cells express a variety of co-stimulatory and co-inhibitory molecules which are activated by specific molecules on antigen-presenting cells. These co-stimulatory and co-inhibitory molecules are recently called immune checkpoint molecules.

**Figure 2 jcm-10-02578-f002:**
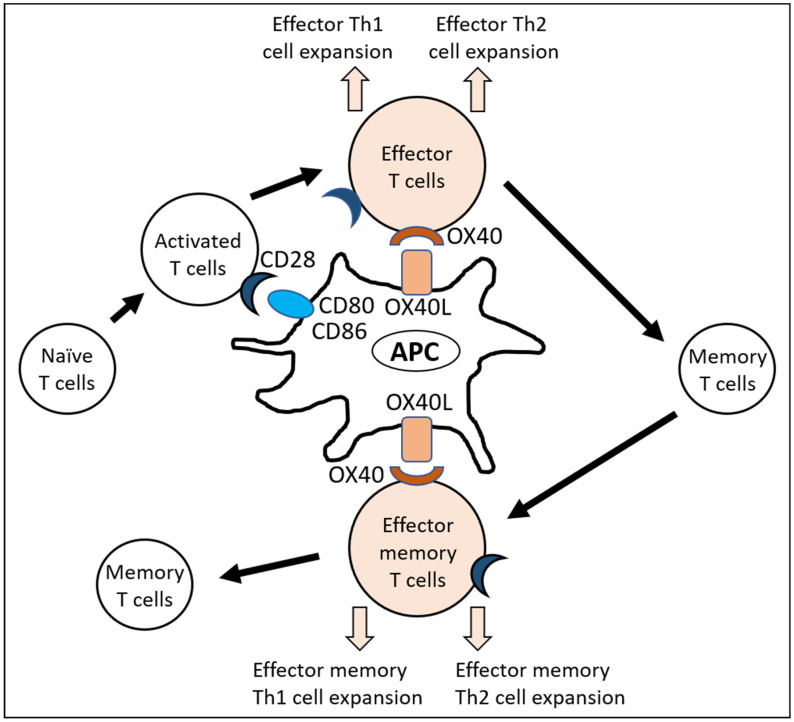
OX40L–OX40 axis induces the expansion of effector Th1 and Th2 cells.Naïve T cells are activated with the help of co-stimulatory molecules such as CD80/CD86 and CD28 ligation. Subsequently, the expansion of effector Th1 and Th2 cells is augmented and sustained by OX40L and OX40 ligation. Resting memory T cells do not express OX40. Upon reactivation, memory T cells become effector memory T cells and express OX40. Ligation of OX40 with OX40L again induces the expansion of effector memory Th1 and Th2 cell expansion. Th1, T helper 1; Th2, T helper 2; APC, antigen-presenting cells.

## Data Availability

Not applicable.

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
