# Peer review of "OX40L–OX40 Signaling in Atopic Dermatitis"

_jcm, 2021, doi:10.3390/jcm10122578_

Round 1
Reviewer 1 Report
This review paper is a comprehensive review of the OX40L-OX40 axis in immune system and allergy. It includes the involvement of the OX40L-OX40 axis in T cell differentiation and activation and its clinical relevance to atopic dermatitis. Through this review article, we can understand the mechanism how the anti-OX40 antibody is effective in atopic dermatitis.
Author Response
Reply to the Reviewer 1
This review paper is a comprehensive review of the OX40L-OX40 axis in immune system and allergy. It includes the involvement of the OX40L-OX40 axis in T cell differentiation and activation and its clinical relevance to atopic dermatitis. Through this review article, we can understand the mechanism how the anti-OX40 antibody is effective in atopic dermatitis.
→ Thank you very much for your encouraging comment. We appreciate your positive evaluation.

Reviewer 2 Report
- The abstract does not indicate what the paper's purpose is or what their goal for the manuscript is
- introduction needs more information and background on current atopic dermatitis. It only briefly mentions checkpoint inhibitors for malignancies/melanoma with very little detail or information of how these are used in atopic dermatitis or what current research is.
- The manuscript has no purpose other than summarizing some information, and not very thoroughly. It could be a review but it is not all-encompassing, is missing information, and therefore is not enough for a complete review paper. I'm not sure what their goal was for this manuscript.
- The language is very basic and the organization is confusing, there needs to be more structure and the information needs to flow better.
Author Response
Reply to the Reviewer 2
The abstract does not indicate what the paper's purpose is or what their goal for the manuscript is
→ Thank you very much for your critical comment. We agree with your comment. According to your comment, we expanded the description of two clinical study in order to better focusing on OX40L-OX40 signaling in atopic dermatitis.
Line 172 to 183
In phase 1 trial of KHK4083 enrolling 22 patients with atopic dermatitis, injection of this antibody on day1, day15 and day 29 ablated OX40-expressing cells and significantly attenuated the symptoms of atopic dermatitis compared to the effects of placebo [62]. Continued improvement in the Eczema Area and Severity Index and Investigator's Global Assessment scores were observed throughout the study until day 155 [62]. A phase 2 clinical study of an anti-OX40 monoclonal antibody (KHK4083) in subjects with moderate-to-severe atopic dermatitis was recently completed (NCT03703102, ClinialTrials.gov) [13]. In this study, 274 patients with moderate to severe atopic dermatitis were enrolled. Although the final results are not published yet, this study also showed progressive improvement in efficacy by continuous KHK4083 administration beyond 16 weeks and the potential for long-term sustained therapeutic effect after the completion of KHK4083 treatment [13].
introduction needs more information and background on current atopic dermatitis. It only briefly mentions checkpoint inhibitors for malignancies/melanoma with very little detail or information of how these are used in atopic dermatitis or what current research is.
→ We agree with your comment. But the article is an Invited short commentary paper on OX40L-OX40 signaling in atopic dermatitis, we first introduced OX40L-OX40 checkpoint and moved on atopic dermatitis. We added the following mouse study to show the feasibility of inhibiting OX40L-OX40 axis to improve TSLP-mediated atopic inflammation. We hope your kind understanding and consideration.
Line 138 to 139
“Of note, TSLP-induced atopic inflammation is prevented by OX40L-specific blocking antibody [48].”
The manuscript has no purpose other than summarizing some information, and not very thoroughly. It could be a review but it is not all-encompassing, is missing information, and therefore is not enough for a complete review paper. I'm not sure what their goal was for this manuscript.
→ Thank you very much for your critical comment. We agree with your comment. According to your comment, we expanded the description of two clinical study in order to better focusing on OX40L-OX40 signaling in atopic dermatitis.
Line 172 to 183
In phase 1 trial of KHK4083 enrolling 22 patients with atopic dermatitis, injection of this antibody on day1, day15 and day 29 ablated OX40-expressing cells and significantly attenuated the symptoms of atopic dermatitis compared to the effects of placebo [62]. Continued improvement in the Eczema Area and Severity Index and Investigator's Global Assessment scores were observed throughout the study until day 155 [62]. A phase 2 clinical study of an anti-OX40 monoclonal antibody (KHK4083) in subjects with moderate-to-severe atopic dermatitis was recently completed (NCT03703102, ClinialTrials.gov) [13]. In this study, 274 patients with moderate to severe atopic dermatitis were enrolled. Although the final results are not published yet, this study also showed progressive improvement in efficacy by continuous KHK4083 administration beyond 16 weeks and the potential for long-term sustained therapeutic effect after the completion of KHK4083 treatment [13].
The language is very basic and the organization is confusing, there needs to be more structure and the information needs to flow better.
→ We agree with your comment. But the article is an Invited short commentary paper on OX40L-OX40 signaling in atopic dermatitis, we first introduced OX40L-OX40 checkpoint and moved on atopic dermatitis. We added the following mouse study to show the feasibility of inhibiting OX40L-OX40 axis to improve TSLP-mediated atopic inflammation. We hope your kind understanding and consideration.
Line 138 to 139
“Of note, TSLP-induced atopic inflammation is prevented by OX40L-specific blocking antibody [48].”
Thank you very much again for your critical comments.
We hope the revised article is now suitable for publication in JCM.

Reviewer 3 Report
This review paper focused on OX40L-OX40signaling in atopic dermatitis.
The review is compact and comprehensive, and seems to be very worthwhile for the readers.
It might be better to add mouse paper that showed feasibility of inhibiting OX40L-OX40 axis to improve atopic dermatitis. One such candidate is “Seshasayee D et al J Clin Invest 117: 3868, 2007”.
As the title of this paper is “OX40L-OX40 signaling in atopic dermatitis”, it might better to describe a bit more about the two clinical studies.
Author Response
Reply to the reviewer 3
This review paper focused on OX40L-OX40signaling in atopic dermatitis.
The review is compact and comprehensive, and seems to be very worthwhile for the readers.
→ Thank you very much for your encouraging comment. We appreciate your positive evaluation.
It might be better to add mouse paper that showed feasibility of inhibiting OX40L-OX40 axis to improve atopic dermatitis. One such candidate is “Seshasayee D et al J Clin Invest 117: 3868, 2007”.
→ Thank you very much for your helpful comment. According to your comment, we added the following sentence by adding Seshasayee’s important article.
Line 138 to 139
“Of note, TSLP-induced atopic inflammation is prevented by OX40L-specific blocking antibody [48].”
As the title of this paper is “OX40L-OX40 signaling in atopic dermatitis”, it might better to describe a bit more about the two clinical studies.
→ Thank you very much for your helpful comment. According to your comment, we expanded the description of two clinical study.
Line 172 to 183
In phase 1 trial of KHK4083 enrolling 22 patients with atopic dermatitis, injection of this antibody on day1, day15 and day 29 ablated OX40-expressing cells and significantly attenuated the symptoms of atopic dermatitis compared to the effects of placebo [62]. Continued improvement in the Eczema Area and Severity Index and Investigator's Global Assessment scores were observed throughout the study until day 155 [62]. A phase 2 clinical study of an anti-OX40 monoclonal antibody (KHK4083) in subjects with moderate-to-severe atopic dermatitis was recently completed (NCT03703102, ClinialTrials.gov) [13]. In this study, 274 patients with moderate to severe atopic dermatitis were enrolled. Although the final results are not published yet, this study also showed progressive improvement in efficacy by continuous KHK4083 administration beyond 16 weeks and the potential for long-term sustained therapeutic effect after the completion of KHK4083 treatment [13].
Thank you very much again for your critical comments.
We hope the revised article is now suitable for publication in JCM.

Round 2
Reviewer 2 Report
I believe the responses and edits were appropriate for this manuscript.